# The Riga East University Hospital Stroke Registry—An Analysis of 4915 Consecutive Patients with Acute Stroke

**DOI:** 10.3390/medicina57060632

**Published:** 2021-06-18

**Authors:** Guntis Karelis, Madara Micule, Evija Klavina, Iveta Haritoncenko, Ilga Kikule, Biruta Tilgale, Inese Polaka

**Affiliations:** 1Stroke Unit, Neurovascular Department, Riga East University Hospital, 1038 Riga, Latvia; dr.mmicule@gmail.com (M.M.); dr.e.klavina@gmail.com (E.K.); iveta.hari@gmail.com (I.H.); ilga.kikule@gmail.com (I.K.); biruta.tilgale@gmail.com (B.T.); 2Infectology Department, Rīga Stradiņš University, 1007 Riga, Latvia; 3Neurology and Neurosurgery Department, Rīga Stradiņš University, 1007 Riga, Latvia; 4Institute of Clinical and Preventive Medicine, University of Latvia, 1586 Riga, Latvia; inese.polaka@gmail.com

**Keywords:** stroke, stroke data, transient ischemic attack, intracerebral hematoma, thrombolysis, quality improvement, stroke registry

## Abstract

*Background and Objectives*: A hospital-based stroke registry is a useful tool for systematic analyses of the epidemiology, clinical characteristics, and natural course of stroke. Analyses of stroke registry data can provide information that can be used by health services to improve the quality of care for patients with this disease. *Materials and Methods*: Data were collected from the Riga East University Hospital (REUH) Stroke Registry in order to evaluate the etiology, risk factors, clinical manifestations, treatment, functional outcomes, and other relevant data for acute stroke during the period 2016–2020. *Results*: During a five-year period, 4915 patients (3039 females and 1876 males) with acute stroke were registered in the REUH Stroke Registry. The causative factors of stroke were cardioembolism (45.7%), atherosclerosis (29.9%), lacunar stroke (5.3%), stroke of undetermined etiology (1.2%), and stroke of other determined causes (1.2%). The most frequent localizations of intracerebral hemorrhage were subcortical (40.0%), lobar (18.9%), and brainstem (9.3%). The most prevalent risk factors for stroke were hypertension (88.8%), congestive heart failure (71.2%), dyslipidemia (46.7%), and atrial fibrillation (44.2%). In addition, 1018 (20.7%) patients were receiving antiplatelet drugs, 574 (11.7%) were taking statins, and 382 (7.7%) were taking anticoagulants. At discharge, 35.5% of the patients were completely independent (mRS (modified Rankin Scale) score: 0–2), while 49.5% required some form of assistance (mRS score: 3–5). The intrahospital mortality rate was 13.7%, although it was higher in the hemorrhage group (30.9%). *Conclusions*: Our stroke registry data are comparable to those of other major registries. Analysis of stroke registry data is important for improving stroke care and obtaining additional information for stroke studies.

## 1. Introduction

Every year, strokes affect 16 million people for the first time and cause approximately 5.7 million deaths worldwide. One third of strokes are attributable to intracerebral or subarachnoid hemorrhage, while two thirds are caused by cerebral ischemia [1,2]. Stroke is one of the leading causes of death and long-term disability worldwide [3]. The trends of change in demographic and vascular risk factors indicate that a significant increase in the burden of cerebrovascular diseases in the future is likely. By 2025, 1.5 million European citizens are expected to experience a stroke event each year [4,5]. In 2015, we estimated the overall cost of stroke to the European Union to be €45 billion. This total includes the cost of healthcare services, medicines used to treat stroke, and missed days of work [6,7]. In Latvia, around 7000–8000 people have a stroke annually, and the lifetime prevalence of stroke is 3% [8,9,10]. By analyzing data from stroke registries, healthcare practitioners can monitor changes in the epidemiology of stroke, the clinical course of the disease, and the effectiveness of treatment over time.

## 2. Materials and Methods

In 2016, we initiated a stroke registry at the Riga East University Hospital (REUH) and data were retrospectively obtained and analyzed. The REUH Stroke Registry contains information for all hospitalized patients who fulfill the World Health Organization International Classification of Diseases 10th edition (ICD-10) criteria for an acute stroke [11].

### 2.1. Study Population

This study utilized the data collected for all patients admitted to the Neurovascular Department of the Riga East University Hospital from December 2016 to January 2020 who were diagnosed with acute stroke or transient ischemic attack (TIA), regardless of the time of symptom onset. We included in the stroke registry patients aged ≥18 years with both first-time stroke and recurrent stroke according to ICD-10. A total of 4915 patients were included in this study.

### 2.2. Measures and Definitions

Demographic information and the date and duration of hospitalization were recorded for each patient. The stroke registry also holds information about the frequency of stroke subtypes, risk factors, comorbidities, medications used in prior hospitalization, intrahospital complications, treatment, and functional outcomes.

The time of stroke onset is not always known, so there was an option to mark “wake-up stroke” or “stroke of unknown duration”. For patients with acute ischemic stroke (IS) in the “time window”, the time from hospitalization until the beginning of treatment (intravenous thrombolysis and/or thrombectomy), i.e., “door-to-needle” time, was recorded.

All acute IS subtypes were classified using the modified Trial of ORG 10172 in Acute Stroke Treatment (TOAST) criteria [12]. If a patient was diagnosed with spontaneous hemorrhage, it was subdivided according to its cause: hypertensive, vascular malformation or others, or subarachnoid hemorrhage. There was also an option to note TIA.

Intrahospital complications were defined as non-neurological complications that occurred during the hospitalization and required an intervention. Other data were collected related to medical treatment during hospitalization, stroke prophylaxis, and rehabilitation.

### 2.3. Clinical Course and Outcome

Each patient was evaluated at admission to hospital, during the patient’s hospital stay, and at the time of discharge, using the National Institute of Health Stroke Scale (NIHSS) and the modified Rankin Scale (mRS) to measure the degree of disability or dependence in daily activities before and after stroke.

### 2.4. Statistical Analysis

The descriptive statistical values used were counts and ratios for categorical factors and the median for continuous factors. Groups were compared using a chi-square test in SPSS (IBM Corp. Released 2011. IBM SPSS Statistics for Windows, Version 20.0. Armonk, NY: IBM Corp). A *p* value of <0.05 was considered statistically significant.

## 3. Results

### 3.1. General Characteristics

Between December 2016 and January 2020, 4915 events were registered in the stroke registry. The patients’ characteristics are summarized in Table 1. The median age was 76 years and 38.2% of the patients were male. A female predominance was noted in all of the stroke types.

The incidence of TIA, IS, and hemorrhagic stroke (HS) and the frequency of the stroke subtypes (TOAST classification) are shown in Table 2. The TIA group had a significantly higher proportion of women than the non-TIA group (66.8% vs. 61.4%, respectively; *p* = 0.047). The IS group did not show a statistically significant difference in gender compared to the other groups. By contrast, the HS group had a significantly higher proportion of men than the non-HS group (45.6% vs. 37.3%, respectively; *p* < 0.001).

The frequency of the cerebral hemorrhage subtypes according to hemorrhage localization is shown in Table 3.

Table 4 shows the incidence of the main risk factors, comorbidities, and previous medications in the patient group. The most prevalent risk factor was hypertension (88.8%), followed by congestive heart failure (71.2%) and dyslipidemia (46.7%). The data show that 1356 (27.6%) patients had experienced a prior stroke, 1018 (20.7%) were taking antiplatelet drugs, 574 (11.7%) were receiving statins, and 382 (7.7%) had received anticoagulants.

Brain computed tomography (CT) scans were obtained from all of the patients at admission to the emergency unit. During hospitalization, 15.3% (*n* = 753) of the patients underwent CT angiography, which was performed more often in the HS group than the other groups (*p* < 0.01), while 13.1% (*n* = 643) underwent magnetic resonance imaging, which was performed more often in the IS group than in the HS group (*p* < 0.05). For most patients, the inpatient period involved no complications (84.9%). The most common complications were the decompensation of congestive heart failure (4.9%), pneumonia (4.0%), and severe kidney failure (2.2%) (Table 5).

There were 2488 (72.5%) events with a known precise onset of stroke, and the median interval between the “last well time” and hospital admission for these events was 2 h and 9 min (Table 6).

### 3.2. Ischemic Stroke

The most commonly affected area in IS was the middle cerebral artery (3226 events, 65.5% of the total), followed by the posterior cerebral artery (9.5%) and basilar artery (6.1%). When the location was distributed by IS subtype, significantly more strokes affecting the middle cerebral artery were cardioembolic (*p* < 0.01) than the other subtypes. Cardioembolic stroke was the most frequent type of stroke overall (Table 2). The distribution of IS subtypes differed by age group. All of the subtypes of cerebral infarction, except for stroke of another determined origin, were more common in those aged over 60 years. The highest incidence in all stroke subtypes was observed in those aged over 80 years. A total of 723 IS events in the stroke registry (17.8%) had received intravenous thrombolysis. The median (IQR) “door-to-needle” time was 42 min (32–57 min). Thrombectomy was performed on 47 patients (1.2%). The most common localization was the middle cerebral artery M1 segment (*n* = 23), followed by the intracranial part of the internal carotid artery (*n* = 16).

### 3.3. Cerebral Hemorrhage

Cerebral hemorrhage was found in 502 patients. Most of the events involved hypertensive intracerebral hemorrhage (ICH) (ICH; 85.0%), while 2.8% were ICH caused by vascular abnormalities and 12.2% due to other causes. In the majority of cases, the ICH volume was below 20 mL. Subarachnoid hemorrhage was found in 8.17% of the patients, while intraventricular hemorrhage was observed in 14.38% of patients (*n* = 88). The most common localizations of ICH were subcortical (40.04%), lobar (18.94%), and brainstem (9.31%) (Table 3). As expected, subcortical hemorrhage was associated with hypertension more often than the other groups (*p* < 0.01), while lobar hemorrhage was more often caused by vascular abnormalities and other causes (*p* < 0.01). Twenty-eight (5.6%) patients were taking oral anticoagulants at hemorrhage onset.

### 3.4. Stroke Outcome

Before stroke onset, 3401 patients (69.2%) could carry out all usual duties and activities. The median NIHSS score was 7. As expected, the median NIHSS score was greater in the cerebral hemorrhage group (median score = 12) (Table 6). The median duration of hospitalization was seven days. At discharge, 35.5% of the stroke patients were completely independent (mRS scores of 0–2) with good outcomes. However, 49.5% of the patients still needed some form of assistance (mRS scores of 3–5) for daily living activities due to various degrees of disability. In these cases, a rehabilitation team provided an initial assessment to all patients when their rehabilitation began, and rehabilitation continued on an outpatient basis. After discharge, 65.2% of the patients returned to their previous place of living, while 11.3% were transferred to nursing facilities. The overall intrahospital mortality rate was 13.7%, with a higher rate in the hemorrhage subgroup (30.9%; Table 6).

## 4. Discussion

Hospital-based stroke registries provide useful additional information on the subtypes, etiological mechanisms, precise clinical patterns, and outcomes of stroke. The information gathered enables stroke unit teams to assess the quality of the medical care provided.

The median age of the patients in this study was 76 years, which is older than that described by other studies in which the average age of stroke patients was 61–72 years [13,14,15,16]. Unlike in other studies that found the incidence of stroke was higher in men, our data confirmed a female-dominant pattern in stroke [13,14,15,16]. However, other studies have shown that the incidence of stroke increases in women at an older age, thereby changing the pattern of higher incidence of male stroke at younger ages [17]. Thus, given that the average age of the patients in our study was 76 years, it is understandable that females were dominant in our study. Among the 4915 acute stroke patients, IS was observed in 83%. HS only occurred in 10% of patients, and this figure is similar to those reported in other stroke studies, which reported ranges between 8% and 25% [13,14,15]. Due to the high recurrence rate of cerebrovascular events, particularly IS, we also collected data from patients with recurrent stroke. Previous reports have indicated that 30% of IS patients and 19% of HS patients have recurrent stroke [18]. Including these patients provides a more realistic view of acute stroke patients, their functional status before and after hospitalization, and the effectiveness of secondary stroke prevention.

The most common etiologies for IS were cardioembolism and atherothrombosis. Although these results differ slightly from the findings of other studies, they are still broadly comparable with data from other studies [13,14,15,19,20]. Cardioembolic stroke accounted for 20–31% of acute ischemic infarctions in those previous studies; while the median age of the patients in our study was higher, cardiac embolism is known to be more common with increasing age, and a characteristic of our dataset was a higher median age of patients [1,21,22].

The most common localizations of ICH were subcortical (40.04%) and lobar (18.94%). As expected, subcortical hemorrhages more often presented with a hypertensive etiology. The most frequent topography of hemorrhages differed from the reports of other studies, where the most common topography was lobar localization [13,14,23].

Overall, our results related to the prevalence of stroke risk factors were similar to those reported by other stroke registries. The most common comorbidities were hypertension, atrial fibrillation, diabetes, cardiovascular disease (congestive heart failure and coronary artery disease), chronic kidney disease, and malignancy. Increasing age is known to be associated with a higher number of comorbidities [24]. However, we observed that among our patients, the percentage of patients with high blood pressure was, in general, higher (88%) than that reported in other studies. The second most common risk factor was carotid artery stenosis (34.7%, with 18.3% showing stenosis greater than 50%). The incidence of atrial fibrillation in our study (44.2%) was higher than that reported previously (18–30%) [25,26]. Compared to a Swedish study, our study showed differing incidences of various comorbidities: chronic kidney disease (11.6% vs. 2.2%, respectively), diabetes (16.8% vs. 19.8%), congestive heart failure (71.2% vs. 6.9%), hypertension (88.8% vs. 58.4%), and atrial fibrillation (44.2% vs. 24.4%) [25]. The most frequently noted risk factors in our data compared to those from a Greek cohort were dyslipidemia (46.7% vs. 67%, respectively), smoking (9.0% vs. 22.5%), and alcohol abuse (3.2% vs. 9.2%) [4]. All of the abovementioned stroke registries include TIA as a risk factor for IS. As TIA and brain infarction have the same pathophysiology and mechanism and share the same prevention strategies, it would be acceptable to consider the history of brain infarction as a stroke risk factor [13,14,15,16].

A significant proportion of patients were treated with antihypertensive drugs, oral antidiabetics, and antiaggregants because of arterial hypertension, diabetes, and prior stroke. However, the proportion of patients who were receiving anticoagulants and statins was quite low (7.7% and 11.8%, respectively) [1]. Our data indicate a need to improve prevention and identify patients not treated with antihypertensive drugs, oral antidiabetics or insulin, statins, antiaggregants, and anticoagulants to reduce the conventional risk factors of acute IS [27].

The most common complications that occurred in hospital were pneumonia (5.1%), followed by decompensation of congestive heart failure (4.9%) and severe kidney failure (2.2%). The reported in-hospital complication rates differed compared to those described in other studies: pneumonia (5.1% vs. 2.3–13.9%) and urinary tract infection (1.7% vs. 3.6–19%) [27,28,29,30]. The results from other reported studies may have resulted from differences in the standardization of complications, definition of complications, selection of patients, patient cohort size, time of evaluation, and follow-up time [28,30].

The proportion of patients who arrived at the hospital in the appropriate 4.5 h time window from stroke onset and received thrombolytic therapy with tPA was 17.8%. This result is for the data collected over the last five years, but over the last two years, the proportion of patients who received tPA in our hospital rose to 27%. The average “door-to-needle” time was 42 min, which corresponds to the recommendations of the stroke treatment guidelines [31].

On assessing the severity of stroke according to the NIHSS scale, we found higher rates of NIHSS scores in the HS group (mean = 12) than in patients with IS (mean = 8). Notably, mortality was much higher in the HS group (30.9%) than in the IS group (12.8%). Patients with HS also had a higher percentage of disability (mRS score = 3–5) identified in assessment of patient functional status at discharge. Similar trends were observed in other studies, which showed both more severe strokes and relatively higher mortality in the HS group [15,17,23]. However, the overall mortality in stroke patients is highly variable across studies [15,23].

Given the mean age of the stroke patients in our study, it was not surprising to find that only 69.2% of the patients (mRS score = 0–1) had a good functional status before hospitalization. Thus, 30.8% of the patients had varying degrees of disability before acute stroke (mRS score ≥ 2). At discharge, 35.5% of the stroke patients were generally able to take care of themselves (mRS score = 0–2) with good overall outcomes. However, almost half of the patients (49.5%) needed some form of assistance (mRS score = 3–5) for activities of daily living. It should be noted that in our study, 17% of TIA patients had a history of cerebral infarction, and only 83% of the patients in this group had mRS scores of 0–1 before hospitalization.

Our study has some limitations that should be explained. The first is related to our stroke registry. The observational data were obtained from a single hospital’s stroke registry and may not reflect the data that would be obtained from a national stroke registry. An ideal stroke registry would be nationwide to increase representativeness and to avoid selection biases [32,33]. The second limitation involves missing data; certain data were missing or were not included in analyses due to data entry errors. The third limitation is the retrospective nature of the stroke registry, which may have resulted in variations in the clinical scores reported by investigators. The fourth limitation is the absence of more detailed data on socioeconomic status and lifestyle factors, which precluded the development of hypotheses regarding the influence of these factors on stroke morbidity and mortality.

Based on the findings of our study, important data on stroke prevention and treatment were collected. We conclude that additional educational work is needed for both the public and doctors to emphasize the importance of stroke risk factors and to help reduce stroke incidence. Moreover, we found that some improvements are needed in the hospital phase of treatment. On the basis of the findings obtained by this study, we have started work on the establishment of a national stroke registry.

The future directions of using our database—we consider making our Stroke register public to learn and query the data, create a National Stroke register to collect data from all hospitals in our country, and participate in other multi-center databases. We have improved our Stroke registry already; 2021, added additional information, and started to collect new data, for example, to establish the role of other possible stroke sequelae inflammatory biomarkers and its clinical relationship between systemic markers of inflammation and cognition and dementia.

## 5. Conclusions

The stroke registry is a useful source of information that can be used to enhance the quality of stroke care and reduce the burden of stroke in Latvia. Improving the quality of the components of the stroke registry will lead to improved acute stroke care. The findings of this study indicate that there has been a marked improvement in the quality of stroke care at the Riga East University Hospital over time, which is comparable to that achieved in other countries.

## Figures and Tables

**Table 1 medicina-57-00632-t001:** Characteristics of the patients with transient ischemic attack (TIA), ischemic stroke (IS), and hemorrhagic stroke (HS).

	Total *n* = 4915	TIA *n* = 355 (7%)	IS *n* = 4058 (83%)	HS *n* = 502 (10%)
	*n* (%)	*n* (%)	*n* (%)	*n* (%)
Male	1876 (38.2)	118 (33.2)	1529 (37.7)	229 (45.6)
Female	3039 (61.8)	237 (66.8)	2529 (62.3)	273 (54.4)
Age, years (median InterQuartile Range—IQR)	76 (66–83)	75 (65–82)	77 (67–83)	70 (59–75)

TIA—transient ischemic attack, IS—ischemic stroke, HS—hemorrhagic stroke, IQR—InterQuartile Range.

**Table 2 medicina-57-00632-t002:** Incidence of TIA, IS, and HS and frequency of stroke subtypes using the TOAST classification.

Characteristic	*n* (%)	Cerebrovascular Event	*n* (%)
Large-artery atherosclerosis	1213 (29.9)	TIA	355 (7)
Lacunar stroke	215 (5.3)	IS	4058 (83)
Cardioembolic stroke	1852 (45.7)	HS	502 (10)
Stroke of other determined causes	51 (1.2)	Total	4915
Stroke of undetermined etiologies	727 (17.9)		

**Table 3 medicina-57-00632-t003:** Frequency of cerebral hemorrhage subtypes according to hemorrhage localization.

Hemorrhage Localization (*n* = 502)	*n*	%
Hemisphere cortical	27	4.42
Hemisphere subcortical	245	40.04
Hemisphere lobar	116	18.94
Cerebellum	29	4.74
Mesencephalon	12	1.96
Pons	38	6.21
Medulla oblongata	7	1.14
Intraventricular	88	14.38
SAH	50	8.17

SAH: subarachnoid hemorrhage.

**Table 4 medicina-57-00632-t004:** Comparison of frequency by chi-square test of the risk factors, comorbidities, and medications used before hospitalization between TIA, IS, and HS.

Cerebrovascular Event		TIA		IS		HS		Total	
Frequency		*n*	%	*n*	%	*n*	%	*n*	%
Risk factors									
Hypertension		302	85.1 ^a^	3602	88.8	461	91.8	4365	88.8
Previous cerebrovascular event	Previous TIAs	43	12.1 ^a^	47	1.2	4	0.8	94	1.9
	Previous IS	61	17.2	1200	29.6 ^b^	95	18.9	1356	27.6
Carotid artery stenosis		137	38.6	1523	37.6	44	8.8	1704	34.7
Smoking		27	7.6	372	9.2	42	8.4	441	9.0
Dyslipidemia		193	54.4 ^a^	2015	49.7 ^b^	88	17.5	2296	46.7
Alcohol abuse		8	2.3	108	2.7	39	7.8	155	3.2
Overweight		58	16.3	837	20.6	73	14.5	968	19.7
Selected comorbidities									
Valvular heart disease		33	9.3	424	10.4	45	9.0	502	10.2
Atrial fibrillation		109	30.7	1972	48.6	89	17.8	2170	44.2
Diabetes mellitus		58	16.3	716	17.6	52	10.4	826	16.8
Coronary artery disease		90	25.4 ^a^	1295	31.9 ^b^	92	18.3	1477	30.1
Congestive heart failure		195	54.9	3018	74.4 ^b^	287	57.2	3500	71.2
Malignancy		16	4.5	207	5.1	18	3.6	241	4.9
Thyroid pathology		18	5.1	242	6.0	17	3.4	277	5.6
Chronic kidney disease		27	7.6	496	12.2	47	9.4	570	11.6
Medication group									
Antihypertensives		244	68.7	2933	72.3	315	62.7	3492	71.0
Antiarrhythmic drugs		22	6.2	248	6.1	11	2.2	281	5.7
Lipid-lowering drugs		67	18.9	478	11.8	29	5.8	574	11.7
Antiplatelets		105	29.6	868	21.3	45.0	9.0	1018	20.7
Oral antidiabetics		28	7.9	289	7.1	17	3.4	334	6.8
Insulin		10	2.8	110	2.7	7	1.4	127	2.6
Oral anticoagulants		40	11.3	314	7.7	28	5.6	382	7.7

TIA: transient ischemic attack; IS: ischemic stroke; HS: hemorrhagic stroke. Chi-square test ^a^ *p* < 0.01 (comparison between TIA and HS). ^b^ *p* < 0.01 (comparison between IS and HS).

**Table 5 medicina-57-00632-t005:** Stroke complications during hospitalization.

Complication	TIA		IS		HS		Total	
	Count	*n*%	Count	*n*%	Count	*n*%	Count	*n*%
No complication during hospitalization	341	96.1%	3401	83.8%	429	85.5%	4171	84.9%
Acute myocardial infarction	2	0.6%	50	1.2%	3	0.6%	55	1.1%
Heart failure decompensation	4	1.1%	229	5.6%	10	2.0%	243	4.9%
Severe kidney failure	4	1.1%	89	2.2%	13	2.6%	106	2.2%
COPD/bronchial asthma exacerbation	1	0.3%	23	0.6%	0	0.0%	24	0.5%
Pneumonia	0	0.0%	211	5.2%	40	8.0%	251	5.1%
Urinary tract infection	0	0.0%	78	1.9%	5	1.0%	83	1.7%
I/V catheter infection	0	0.0%	8	0.2%	1	0.2%	9	0.2%
Bedsores	0	0.0%	49	1.2%	3	0.6%	52	1.1%
GIT bleeding	0	0.0%	8	0.2%	1	0.2%	9	0.2%
Deep vein/pulmonary artery thrombosis	1	0.3%	53	1.3%	3	0.6%	57	1.2%
Other hospital infections	0	0.0%	52	1.3%	8	1.6%	60	1.2%

COPD: chronic obstructive pulmonary disease; I/V: intravenous; GIT: gastrointestinal tract.

**Table 6 medicina-57-00632-t006:** Duration of hospitalization, stroke severity on the National Institute of Health Stroke Scale (NIHSS), and functional outcome on the mRS.

		TIA *n* (%)	IS *n* (%)	HS *n* (%)	Total *n* (%)
		Median (IQR)	Median (IQR)	Median (IQR)	Median (IQR)
Last well time to admission interval, hours		1:48:29 (1:11:59–3:47:59)	2:16:00 (1:18:00–6:05:59)	1:44:59 (1:06:59–4:15:00)	2:09:00 (1:15:59–5:39:59)
NIHSS score at admission		0	8 (4–15)	12 (4–20)	7 (4–15)
NIHSS score at discharge		0	4 (2–9)	4 (2–11)	3 (1–8)
mRS before hospitalization					
0		242 (68.2)	1895 (46.7)	274 (54.6)	2411 (49.1)
1		50 (14.1)	836 (20.6)	104 (20.7)	990 (20.1)
2		22 (6.2)	472 (11.6)	38 (7.6)	532 (10.8)
3		19 (5.4)	356 (8.8)	28 (5.6)	403 (8.2)
4		3 (0.8)	208 (5.1)	21 (4.2)	232 (4.7)
5		0 (0.0)	72 (1.8)	11 (2.2)	83 (1.7)
Unknown		19 (5.4)	219 (5.4)	26 (5.2)	264 (5.4)
mRS at discharge					
0		248 (69.9)	226 (5.6)	19 (3.8)	493 (10.0)
1		53 (14.9)	568 (14.0)	60 (12.0)	681 (13.9)
2		19 (5.4)	517 (12.7)	32 (6.4)	568 (11.6)
3		22 (6.2)	680 (16.8)	53 (10.6)	755 (15.4)
4		4 (1.1)	839 (20.7)	107 (21.3)	950 (19.3)
5		2 (0.6)	655 (16.1)	72 (14.3)	729 (14.8)
Death		0	520 (12.8)	155 (30.9)	675 (13.7)
Unknown		7 (2.0)	53 (1.3)	4 (0.8)	64 (1.3)
Duration of hospitalization, days		5 (4–7)	8 (6–9)	7 (5–9)	7 (6–9)

NIHSS—National Institute of Health Stroke Scale.

## Data Availability

Data are available upon request due to restrictions (ethical). The data presented in this study are available on request from the corresponding author. The data are not publicly available due to localization in the hospital Stroke Registry.

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
