# Peer review of "The Riga East University Hospital Stroke Registry—An Analysis of 4915 Consecutive Patients with Acute Stroke"

_medicina, 2021, doi:10.3390/medicina57060632_

Round 1
Reviewer 1 Report
Hospital-based stroke registry is a useful tool for systematic analyses of the epidemiology, clinical characteristics, and natural rehabilitation after stroke. Data were collected from the Riga East University Hospital (REUH) Stroke Registry in order to evaluate the etiology, risk factors, clinical manifestations, treatment, functional outcomes, and other relevant data for acute stroke during the period 2016–2020. However, it is very important to outline the future directions of using the data base, for example investigate the clinical relationship between systemic marker of inflammation and cognition and dementia (see, DOI: 10.1038/SREP13281)
Reviewer 2 Report
This is a single center retrospective review and registry.There are not many of these available outside major medical centers in the western world / this offers insight into a different setting. Some of their findings - including the female to male incidence - are unique. Smaller center.More robust data collection.The paper is well written. This is a database and retrospective review and they address the main question posed.
Very well-done manuscript, and an important contribution to the literature as there are very few single center databases like this one available outside major Western medical centers. You should consider making the database online for others to learn and query the data, or participate in other multi-center databases such as the STAR registry: https://medicine.musc.edu/departments/neurosurgery/star/sites. I am certain they (and others) would welcome your input.
Round 2
Reviewer 1 Report
The authors did not address my observations at the DISCUSSION
Author Response
Please see the attachment.

This manuscript is a resubmission of an earlier submission. The following is a list of the peer review reports and author responses from that submission.
Round 1
Reviewer 1 Report
Hospital-based stroke registries offer the advantage of providing additional information of subtypes of strokes, etiological mechanisms, precise clinical patterns, and outcomes. Additionally, the collected results will allow stroke unit teams to assess the quality of the provided medical care.
Comments
This is a very interesting study shedding some light on nation-specific risk factors for stroke. However, I was surprised to learn that aadvanced age was not a risk factor for stroke (see, .
Also at hemorrhagic stroke a critical inflammatory factor has not been addressed or at least discussed. What about cognition, has it been also assessed? (see, DOI: 10.1038/SREP13281; DOI: 10.1007/S10522-014-9516-1)
Author Response
Thank you for your comments!
We have made english language and style spell checking. We agree with you that age is risk factor for stroke and this fact is reflected in our work. But the critical inflammatory of hemorrhagic stroke and cognition was not analyzed because these data was no collected in database and it was not task of this study as well.

Reviewer 2 Report
In this paper the authors in number of 14!!! present data from Stroke Registry.
The original research should focus on a special issue from Stroke Registry not only show data that is known and fully consistent from many years.
This paper brings no novelties to the current knowledge.
Please also check the criteria for coauthorships- I understand that a lot of persons were involved in Stroke Registry preparation, but presented data with only one statistical method (chi-sguare) may be done by max. 3-4 researchers.
Please use tha data form Stroke Registry in the proper scientific manner- to present ideas, conceptions and innovations, not only provide general information about stroke.
Author Response
Than you for your comments!
We have revised the number of authors and check the criteria for coauthorships - now there are only 7 authors. We have made English language corrections as well. We have improved the results presentation and in the discussion section we presented and reflected our ideas based on the results that have been obtained.
Round 2
Reviewer 1 Report
Stroke is increasingly recognized as an important cause of cognitive problems and has been implicated in the development of both Alzheimer’s disease and vascular dementia
The authors shall discuss the potentail impact of systemic inflammation and development of dementia after stroke (see, (DOI: 10.1038/SREP13281))
Author Response
Thank you for comments. We analyzed them and taken into account. Our hospital's Stroke Registry was created six years ago. It included certain questions to gather information about the stroke. Of course, it would be interesting and useful to observe other risk factors, for instance - to evaluate cognitive impairment. However, we did not collected such information in our Stroke Registry and it is not possible to perform analysis of these factors currently. We would take this into account and improving our Stroke Registry by obtained additional information in the future.
The English language, style and paper scientific information were edited according to the requirements of the journal and were done by journal editor team. We have attached the Certificate.

Reviewer 2 Report
In contrast to the previous version the authors added only one sentence in Discussion: "Based on the findings of our study, important data on stroke prevention and treatment have been collected. We conclude that additional educational work is needed for both the public and doctors to emphasize the importance of stroke risk factors and help reduce stroke incidence. Moreover, we have found that some improvements are needed
in the hospital phase of treatment. On account of the findings obtained by this study, we have started work on the establishment of a national stroke registry. "....
Additionally, English corrections and reduction in number of coauthors have been made.
These are cosmetic, far insufficient improvements to the previous version.
The main goal of my criticism has not been highlighted and revised.
Author Response
Thank you for comments. We analyzed them and taken into account. Our hospital's Stroke Registry was created six years ago. It included certain questions to gather information about the stroke. Of course, it would be interesting and useful to observe other risk factors as well. We agree that the analysis and presentation of data collected by the Stroke Registry is not a novelty. However, in our case, the information presented in the article is important at our national level to reflect the Stroke Registry data. At the Latvian country level, this is a novelty. That kind of analysis has not been performed in our country before. Our task was not to come up with new ideas, but to analyze improvements in stroke care and treatment in our hospital and country in the future by analyzing the data. Similar papers can be found in different medical databases, which also include Stroke Registry data that reflected the results of the work. They also have no novelty. Therefore, we do not understand your comments about the submission of new ideas, novelty and necessary significant improvements. The collection, analysis and presentation of our data had a different purpose, which I mentioned earlier. Therefore, additional comments were added during the discussion. The English language, style and paper scientific information were edited according to the requirements of the journal and were done by journal editor team. We have attached the Certificate.
